# Development and Validation of the Career and Life Development Hope (CLDH) Scale among Non-Engaged Youth in Hong Kong

**DOI:** 10.3390/ijerph191610283

**Published:** 2022-08-18

**Authors:** Steven Sek-yum Ngai, Chau-kiu Cheung, Lin Wang, Yunjun Li, Yuen-hang Ng, Elly Nga-hin Yu, Winnie Pui-ching Leung

**Affiliations:** 1Department of Social Work, The Chinese University of Hong Kong, Hong Kong, China; 2Department of Social and Behavioral Sciences, City University of Hong Kong, Hong Kong, China

**Keywords:** career and life development hope, non-engaged youth, school-to-work transition, scale validation, career intervention

## Abstract

Increasing career and life development hope (CLDH) is critical for the career and life pursuits of non-engaged youths (NEY) who face various disadvantages in the school-to-work transition, especially considering current challenging labor market conditions and the impacts of the pandemic. Nevertheless, research that explores the assessment of CLDH among NEY is scarce. To address this gap, this study aimed to develop and validate a CLDH measurement instrument. A total of 1998 NEY aged 13–29 years in Hong Kong participated in our study. Exploratory factor analysis of the 20-item CLDH scale suggested a two-factor structure—career and life development pathways (CLDP) and career and life development agency (CLDA)—which accounted for 63.08% of the total variance. The confirmatory factor analysis results show a good model fit (CFI = 0.934, TLI = 0.926, RMSEA = 0.060, 90% CI [0.055, 0.065], SRMR = 0.042) and all the items significantly represented the corresponding sub-constructs. The results also demonstrate a satisfactory internal consistency for all subscales and the full scale (0.89–0.95). Sub-group consistency across subsamples categorized by gender, age, and years of residence in Hong Kong was indicated. Correlations between the CLDH scale and subscales with other career-related and social well-being outcomes (i.e., youth career development competency, career adaptability, civic engagement, social contribution, and social integration) showed good concurrent validity. Our results support that the CLDH scale is a valid and reliable tool for measuring NEY’s hope for career and life development in the Hong Kong context. Theoretical and practical implications of the findings are also discussed.

## 1. Introduction

The current labor market is characterized as dynamic and uncertain, with a high demand for resilience, adaptability, and self-direction [1,2] for both the employed and those in the status of school-to-work transition. Globally, more than 20% of young people were not in employment, education, or training (NEET) in 2019, and young people were more than three times more likely to be unemployed than adults over 25 years old [3]. Likewise, influenced by the COVID-19 pandemic, the unemployment rates among young people aged 15–20 and 20–29 in Hong Kong were 17.2% and 7.7%, respectively, from February to April 2022, whereas the unemployment rate among the general population was 5.5% during the same period [4]. The unfavorable and difficult career environment and the high unemployment rate make gainful employment a greater challenge [5]. This is especially difficult for non-engaged youths (NEY, e.g., school dropouts, unemployed youths, and ethnic minority youths) who face different disadvantages in their school-to-work transitions and are at high risk of becoming NEET youths [6,7]. The characteristics of NEY’s disadvantaged circumstances, such as being discriminated in education and employment and having negative prior experiences during their developmental trajectories, make it difficult for them to be hopeful about achieving their desired goals, resulting in the likelihood of unstable employment, poor health, and behavioral problems [8,9,10,11,12].

Furthermore, career and life development theories stress the importance of young people developing hope [13,14]. It is the perceived ability to successfully follow pathways to desired goals and motivate oneself to use those pathways through agency thinking [15]. Scholars in related disciplines (e.g., vocational psychology and career counseling) propose that hope is also highly relevant for anyone engaged in career and life development pursuits [16,17], especially in the face of the intense competition and uncertainty that characterize the current labor market. Indeed, as evidenced by the topics of numerous international conferences [18] and scholarly publications, hope is increasingly positioned as a key state in positive career and life development [19,20]. In this context, being hopeful in career and life development is especially relevant and important for NEY since their economic disadvantage and lack of motivational strength put them in an even more disadvantaged position, with insufficient knowledge about different career pathways and low educational and vocational aspirations [21,22].

Accordingly, increasing career and life development hope (CLDH)—a positive motivational–cognitive–behavioral state toward achieving career and life development goals—is critical for NEY’s career and life pursuits [16,23,24,25,26,27,28] and helps them to develop a sense of personal agency and control in career- and life-goal-directed behavior [15,25]. Moreover, hope is demonstrated to be associated with positive events that people expect will occur in the future, and it is identified as an important predictor of positive developmental outcomes (e.g., academic achievement, persistence, attachment, interpersonal relationships, and psychological well-being) [15,29,30,31,32,33]. However, there is little research on the development and validation of CLDH instruments for disadvantaged young people. To address the measurement gap, the present study aims to develop the CLDH scale for measuring young people’s motivational–cognitive–behavioral state in terms of the agency and pathways components of hope for career and life development in a sample of NEY in Hong Kong. This scale can enable measurements of young people’s CLDH in research and practice and inform the development of policies and interventions to enhance NEY’s hope for career and life development to help them improve well-being and achieve success.

## 2. Literature Review

### 2.1. Conceptualization of Hope and Theoretical Basis

The concept of hope has received increased attention in youth studies in recent years. Scholars attempt to conceptualize and measure hope to better understand its influence on the development of young people [34]. According to hope theory, hope in young people consists of a motivational–cognitive–behavioral state involving self-related appraisals about one’s abilities to produce workable routes to goals (the pathways component), as well as initiating and maintaining movement toward those goals (the agency component) [15,22,35]. In other words, hope is considered to be a positive motivational–cognitive–behavioral state with two components [15]. The motivational component, agency, refers to the ability to initiate and sustain motivation to achieve desired outcomes. The cognitive–behavioral component, pathways, refers to the perceived cognitive and behavioral ability to generate multiple routes or pathways to achieve desired goals. The two components are thought to reinforce each other during goal pursuit and achievement [15,22,36]. Furthermore, it is worth noting that the conceptualization of hope is distinct from that of self-efficacy [15,37]. Specifically, self-efficacy refers to one’s perceived ability to succeed in given tasks, which tends to be tied very narrowly to situation-specific goals [38]. By contrast, hope pertains more to the generic positive expectation of goal attainment [15] and is concerned with both an individual’s planning (i.e., the pathways component) and motivation (i.e., the agency component) that will lead to goal achievement [39]. Importantly, pathway generation as well as its dynamic interplay with the agency component have been pointed out as critical elements for differentiating hope from self-efficacy [37], which further contributes to our understanding of hope as a dynamic motivational–cognitive–behavioral state that moves an individual toward achieving career and life development goals [26].

Despite the growing recognition of the importance of hope concerning various developmental outcomes, empirical research on hope in the context of career and life development is still in its early stages and remains scarce. Work hope, a domain-specific expression of hope, was extensively studied and found to be positively related to career development variables, such as career planning [40], career self-efficacy beliefs [16], and vocational identity [13]. However, what CLDH is and how it is measured among young people, particularly NEY, has not yet been studied. Because hope is important in active coping and the proactive pursuit of goals [15], more empirical research is needed to develop and validate a scale that measures hope for career and life development among NEY so that we can better understand their psychological condition for career and life development. Because of the growing concern for how hope can change in NEY populations, a better understanding of hope in career and life development may also contribute to improved career intervention and human resource development practice [41,42].

### 2.2. Measurement Gaps

#### 2.2.1. CLDH in a Hong Kong Chinese Context

What may complicate the CLDH of NEY in Hong Kong are the unique sociocultural characteristics of the Hong Kong Chinese context. The competitive labor market conditions concomitant with the dramatic impacts of the COVID-19 pandemic have made young people in Hong Kong a disadvantaged group undergoing difficulties in career and life development [43]. As mentioned in the introduction, the alarmingly high unemployment rate signifies that job opportunities for today’s young people in Hong Kong are decreasing. The attention to career and life development among NEY, who are generally more excluded from economic activities [9], needs refocusing. In particular, compared with the general youth population, NEY are found to be more likely to bounce back and forth between normal employment, atypical employment, unemployment, and school [9,44], and end up with a lower-wage job, poorer working conditions, and less stability [45]. Consequently, NEY are more likely to find it extremely challenging to find stable employment and thereby experience a greater risk of becoming lost, feeling hopeless, and losing motivation [46,47].

Additionally, Hong Kong studies demonstrate that NEY not only face employment difficulties, but also experience a series of emotional, social, and behavioral problems that are strongly associated with the non-engagement status, including mood disorders, anxiety, a lack of confidence, social withdrawal and isolation, learning disability, substance use, and participation in criminal groups [6,48,49]. For NEY, long-term non-engagement and its negative outcomes are likely to lead to hopelessness, which makes the problem even worse. For instance, past research has revealed that NEY tend to lack hope and confidence for the future, including feeling unable to secure a job or losing purpose in life [9,50].

Furthermore, the characteristics of Chinese culture may influence the CLDH of NEY in Hong Kong. In the Chinese context, the endorsement of traditional beliefs plays an important role in shaping individuals’ career and life pursuits, which include, but are not limited to, the fulfillment of social expectations, filial piety, and respect for and submission to authority [51]. Strongly influenced by the expectations of developing a desirable social image and gaining respect from others, Chinese individuals typically value higher achievement in career and life development, and certain jobs that are “high-income” and “high-status” are therefore widely considered “right” career choices [43,52], which may exert negative impacts on NEY’s hope for career and life development. Additionally, Chinese parents might explicitly view higher education as the most promising choice for their children [53] or prematurely discourage young people from considering alternative career and life pathways, especially when those parents’ knowledge of contemporary employment trends are not up to date [54]. As a consequence, while Chinese young people are commonly expected to respect and submit to their parents’ advice and rules instead of following their individual preferences [43], previous research has revealed that NEY often feel unsupported and discouraged by their parents when attempting to explore different or alternative career and life pathways [55]. The incongruence between NEY’s aspirations and social expectations may further increase their level of powerlessness and hopelessness.

Taken together, given the challenging circumstances of the Hong Kong Chinese context for NEY, as well as the significance of hope for career and life development, there is an urgent need to develop a rigorous and valid measurement instrument that can be used by concerned professionals to assess the status of CLDH among NEY and understand and identify the possible needs of NEY for successfully navigating the transitions through education into employment.

#### 2.2.2. Career-Related and Social Well-Being Outcomes among NEY

Aside from a scarcity of research on CLDH measurement instruments in Hong Kong’s unique cultural context, there is also a scarcity of research on the current validity of CLDH measurement instruments. First, empirical support for the positive relationships between CLDH and career-related outcomes for young people—including career adaptability and youth career development competency—are widespread [19,40,56]. For example, Diemer and Blustein [13] discovered in empirical studies that vocational hope might be a particularly important consideration in adolescents’ career development. Similarly, Sung et al. [27] found that the agency aspect of hope—one of the components of the employment hope scale—was positively correlated with both skills and outcomes (e.g., career exploration, person–environment fit, goal setting, and work readiness). Furthermore, Santilli et al. [57] discovered that career adaptability predicted life satisfaction indirectly via the two components of hope: agency and pathways.

Furthermore, social well-being refers to people’s assessment of their social situation and functioning in society and includes a variety of components such as feelings of belonging and the belief that one can contribute to society [58,59,60]. According to the current literature, hope shows positive relationships with social well-being outcomes (e.g., social integration, social contribution, and civic engagement) [61,62]. Specifically, empirical studies have shown that hope is related to higher levels of participation in socially inclusive activities [63]. Furthermore, research has found that hopeful young people are more likely to engage in positive and goal-oriented activities in their communities and society [64]. Callina et al. [61] discovered that optimistic future expectations were positively associated with civic engagement. Although significant, the concurrent validity of CLDH measurement instruments has not been demonstrated. Therefore, this study also seeks to establish the concurrent validity of the CLDH scale by evaluating its association with five career-related and social well-being outcomes: youth career development competency, career adaptability, civic engagement, social contribution, and social integration.

#### 2.2.3. Subgroup Consistency of CLDH Scale

Given the features of similar valid and reliable scales [43], analyzing the subgroup consistency of a newly developed scale might be useful for measuring hope for career and life development among NEY. Specifically, when ensuring that an instrument is usable and valid for different subgroups, it is necessary to utilize critical variables in examinations of the possible variations in applications of the CLDH scale so as to confirm the stability of the measurement instrument. Taken from existing studies, these critical variables may include gender, age, and years of Hong Kong residence [43,65,66,67]. Meanwhile, although empirical research has identified possible differences in how CLDH manifests across the aforementioned three demographic subgroups, subgroup consistency results were not reported in previous versions of hope scales. In this context, another primary goal of the present study is to examine the subgroup consistency of the CLDH scale to determine whether this new scale maintains a good model fit in various demographic subgroups.

## 3. Method

### 3.1. Procedure and Participants

Given this study’s purpose of developing and validating a CLDH scale, the study recruited a sample of 1998 respondents aged 13–29 years (*M* = 18.87, *SD* = 3.284) who had participated in a territory-wide project called CLAP@JC, which aims to foster a sustainable ecosystem by bringing together the education, business, and community sectors to smoothen young people’s transition from school to work. Participants were explicitly informed of the study’s objectives, procedures, and related ethical information. For participants under 18 years, their parents’ or guardians’ signed consent was also obtained. All procedures for the current study were approved by the Survey and Behavioral Research Ethics Committee at a major university in Hong Kong and were found to comply with the ethical standards for research involving human subjects.

Of the 1998 participants, 51.9% were male and 48.1% were female. Approximately 89.2% of the participants were Chinese, and 10.8% were South or Southeast Asian minorities; 84.8% were born in Hong Kong, with a median residence in Hong Kong of 18 years; and 76.4% had achieved senior secondary school or higher education. Regarding their current employment status, most participants (43.1%) were unemployed, while 28.5% were students, 2.5% were homemakers, 2.1% were self-employed or temporarily employed, and 23.8% were in regular employment.

### 3.2. Measures

The structured questionnaire employed in the current study contained two parts. The first part consisted of the CLDH scale that was developed by our research team after making reference to the existing literature [15,22]. A total of 20 items were generated for measuring two potential components, namely career and life development pathways and career and life development agency. A panel of six researchers and 10 social workers with experience in the youth service field was invited to independently proofread and refine the scale to ensure its face value. Our research team considered their suggestions and feedback and revised the items accordingly. Subsequently, a pilot study was conducted with 14 NEY to review the clarity of the proposed items. Their feedback on the clarity of the proposed items was incorporated into the revision of the CLDH scale. The second part included measures of career-related and social well-being outcomes (i.e., youth career development competency, career adaptability, civic engagement, social contribution, and social integration), which were adopted from previous research conducted in local or overseas contexts [43,68,69] for checking the concurrent validity of the CLDH scale. The CLDH scale was translated into Chinese. The semantic equivalence of the translated Chinese version to the original scale was verified through back-translation procedures.

#### 3.2.1. CLDH Scale-Components

Career and life development pathways refer to young people’s appraisals about their ability to generate multiple workable routes or pathways to achieve desired goals related to career and life development [15,22]. The current study developed seven items to assess pathways by asking participants about their appraisals toward their personal ability to perform a list of things over the past month. Sample items included “Had a clear direction and meaningful engagements (e.g., education, employment, or training) in career and life development,” “Had encounters with the business sector to understand the industry,” and “Took initiatives to set career and life development goals and plans.” Responses were scored on a five-point scale, where 5 = Always, 4 = Very often, 3 = Sometimes, 2 = Rarely, and 1 = Never.

Career and life development agency refers to young people’s appraisals about their ability to initiate and execute new movements toward attaining desired career and life goals, as well as sustaining the motivation and activities to achieve those goals [15,22]. The current study developed 13 items to assess agency by asking participants about their appraisals toward their personal ability to perform a list of things over the past month. Sample items included “Was confident to make career choices that suit me,” “Felt that future career and life development will be good,” and “Maintained positive work attitudes and life values.” Responses were scored on a five-point scale, where 5 = Strongly agree, 4 = Agree, 3 = Neutral, 2 = Disagree, and 1 = Strongly Disagree.

#### 3.2.2. Youth Career Development Competency

The youth career development competency (YCDC) scale was added in order to verify the concurrent validity of the CLDH scale. It assesses young people’s ability to navigate transitions from school to obtaining a productive and meaningful career [43,70]. In the present study, the 17-item YCDC scale was adapted from the Youth Career Development Competency Scale [43]. Participants were asked about their level of career development competency over the previous month. Sample items include “Continuously participate in my selected activities and new experiences”, “Verify my interests, competences, and values through daily life self-observations”, and “Cope with the future’s career and life development transitions and changes, and the stress involved”. Responses were scored on a five-point scale, where 5 = Highly confident, 4 = Confident, 3 = Neutral, 2 = Not confident, and 1 = Not confident at all. The Cronbach’s alpha of this scale was 0.965.

#### 3.2.3. Career Adaptability

The career adaptability (CA) scale was used to examine the concurrent validity of the CLDH scale. It measures young people’s readiness and resources to make and implement successful career decisions [70]. In the current study, the 12-item CA scale was adapted from the Career Adapt-Abilities Scale [69]. Participants were asked about their level of career adaptability in relation to current and anticipated tasks in their occupational roles over the previous month. Sample items included “Taking care to do things well” and “Learning new skills.” Responses were scored on a five-point scale, where 5 = Very strong, 4 = Strong, 3 = Neutral, 2 = Not strong, and 1 = Not strong at all. The Cronbach’s alpha of this scale was 0.938.

#### 3.2.4. Civic Engagement

The civic engagement (CE) scale was included to test the concurrent validity of the CLDH scale. It assesses young people’s participation in activities that improve conditions for others or enhance their local community [69]. In the present study, the 6-item CE scale was adapted from the Civic Engagement Scale [69]. Participants were asked to rate their level of participation in community service or other related civic activities over the previous month. Sample items included “Participated in community action project” and “Became an active member in the community.” Responses were scored on a five-point scale, where 5 = Always, 4 = Very often, 3 = Sometimes, 2 = Rarely, and 1 = Never. The Cronbach’s alpha of this scale was 0.956.

#### 3.2.5. Social Contribution

The social contribution (SC) scale was included to check the concurrent validity of the CLDH scale. It measures young people’s deeds, behaviors, or activities undertaken to shape society’s future [69]. In this study, the 5-item SC scale was adapted from the Social Contribution Scale [69]. Participants were asked to rate their level of perceived personal contribution to their local or broader community over the previous month. Sample items included “Helped someone in the community” and “Contributed to the community”. Responses were scored on a five-point scale, where 5 = Always, 4 = Very often, 3 = Sometimes, 2 = Rarely, and 1 = Never. The Cronbach’s alpha of this scale was 0.951.

#### 3.2.6. Social Integration

The social integration (SI) scale was included to verify the concurrent validity of the CLDH scale. It assesses young people’s perceived cohesion or sense of belonging to their social support networks, including a group or a larger community [69]. In the present study, the 4-item SI scale was adapted from the Social Integration Scale [69]. Participants were asked to rate the extent to which they perceived themselves as belonging to different social networks, including a group of friends or a larger community, over the previous month. Sample items included “Thought that I am able to play a role in society” and “Hung out with friends”. Responses were scored on a five-point scale, where 5 = Always, 4 = Very often, 3 = Sometimes, 2 = Rarely, and 1 = Never. The Cronbach’s alpha of this scale was 0.786.

### 3.3. Data Analysis

Statistical analysis involved five steps. First, exploratory factor analysis (EFA) was used. In this step, the whole sample was randomly divided into two subsamples. An EFA was performed on one subsample (*n* = 980) using SPSS 26. Additionally, principal component analysis (PCA) was used to determine the factor structure. The factor eigenvalues were set to at least one [71], and varimax rotation was conducted during the PCA. The Kaiser–Meyer–Olkin (KMO) sampling adequacy and Bartlett’s test were used to examine data factorability [72]. Second, confirmatory factor analysis (CFA) was used. In this step, CFA was performed on the second subsample (*n* = 1018) using Mplus 8.2, drawing on the results of the EFA procedures [73] to determine whether the model data fit the item factor structures. Comparative fit index (CFI ≥ 0.90, acceptable) [74], Tucker–Lewis index (TLI ≥ 0.90, acceptable) [75], root mean square error of approximation (RMSEA ≤ 0.08, acceptable) [76], and standardized root mean square residual (SRMR ≤ 0.08, acceptable) [77] were used to evaluate the model fit. We anticipated that if convergent validity exists, relevant subscale items should converge, respectively, to the factors of career and life development agency and career and life development pathways; moreover, the correlation between the constituent factors of the CLDH scale should be significant and positive [78]. Third, subgroup consistency was validated. In this step, CFA was performed using the three subgroups (divided into pairs): gender (male vs. female), age (<19 years (younger) vs. ≥19 years (older)), and years of residence in Hong Kong (<18 years (shorter) vs. ≥18 years (longer)). For age and years of residence in Hong Kong, each group’s median was used as the respective cut-off. Fourth, internal consistency and reliability were established. In this step, the reliability of the CLDH scale and its subscales was assessed using the two-factor structure described in the preceding factor analyses; Cronbach’s alpha coefficients (≥0.700, acceptable) and composite reliability coefficients (≥0.700, acceptable) were used [79]. The fifth step addressed concurrent validity and discriminant validity. Concurrent validity and discriminant validity were assessed by examining the Pearson’s correlation coefficients between the CLDH scale and measures of career-related outcomes (i.e., career adaptability and youth career development competency) [43,69] and social well-being outcomes (i.e., civic engagement, social contribution, and social integration) [69]. We anticipated that if concurrent validity exists, the CLDH scale should be significantly and positively correlated with the abovementioned measures. We also anticipated that if discriminant validity exists, the relationship between the CLDH scale and the abovementioned measures should not be too strong in that their correlation coefficients should be less than the criterion of 0.700 [80]. The normality of the data was tested before correlation analyses were conducted in order to verify concurrent validity and discriminant validity. Essentially, CLDH, CLDP, and CLDA showed normal distributions, which made them suitable for the analyses. We also performed bootstrapping (number of samples = 5000) at a 95% confidence intervals for significance testing.

## 4. Results

### 4.1. Exploratory Factor Analysis

For the CLDH scale, the EFA analysis revealed that the KMO coefficient was 0.966, and the Chi-square from Bartlett’s test was 0.348 (*p* < 0.001). These results indicate that the data were appropriate for PCA. The PCA revealed that the CLDH scale had a two-factor structure that accounted for 63.08% of the variance. The first factor, “career and life development pathways (CLDP),” consisted of seven items; the factor loadings of these items ranged from 0.592 to 0.785, accounting for 39.94% of the total variance in the scale (see Table 1). The second factor, “career and life development agency (CLDA)”, consisted of thirteen items; the factor loadings of these items ranged from 0.705 to 0.810, accounting for 23.14% of the scale’s total variance (see Table 1).

### 4.2. Confirmatory Factor Analysis and Convergent Validity

The fit indices for the CFA model were examined, and the results yielded a significant Chi-square value (χ^2^ = 794.913, *df* = 169, χ^2^/df = 4.704). The fit index values (CFI = 0.934, TLI = 0.926, RMSEA = 0.060, 90% CI [0.055, 0.065], SRMR = 0.042) indicated that the two-factor model yielded a good fit. The factor loadings related to the two-component model are presented in Figure 1, with factor loadings ranged from 0.610 to 0.841 for CLDP and from 0.653 to 0.845 for CLDA. The CFA also showed that relevant subscale items converged, respectively to the factors of CLDA and CLDP. Moreover, the correlation between CLDP and CLDA was 0.636 (*p* < 0.001), which indicated a good convergent validity of the CLDH scale.

### 4.3. Reliability Estimation

Cronbach’s alpha was used to estimate the internal consistency of the scale and of each subscale. The analysis revealed that the total correlations of the corrected items ranged between 0.614–0.758 for CLDP and 0.657–0.817 for CLDA (see Table 2). The results demonstrate a high consistency among the items within each subscale. The Cronbach’s alphas for CLDP and CLDA were 0.892 and 0.950, respectively, and the Cronbach’s alpha for the total scale was 0.948. Additionally, the study estimated composite reliability. The results show that the composite reliability coefficients of the CLDH scale as well as its CLDP and CLDA subscales were 0.828, 0.893, and 0.950, respectively, which surpass the standard of 0.700 [81]. The results demonstrate that all subscales and the entire CLDH scale had a good reliability, indicating that the internal consistency was satisfactory.

### 4.4. Factorial Validation in Subsample

CFA studies were separately conducted across the subsamples of gender, age, and years of residence in Hong Kong (with the goodness of fit shown in Table 3). Satisfactory results were obtained for the subsamples.

### 4.5. Concurrent Validity and Discriminant Validity

As explained above, the concurrent validity of the CLDH scale was determined using both career-related outcomes—career adaptability and youth career development competency [43,69]—and social well-being outcomes—civic engagement, social contribution, and social integration [69]. The correlation coefficients between the CLDH scale, the CLDH subscales, as well as the career adaptability, youth career development competency, civic engagement, social contribution, and social integration scales were calculated in order to confirm the CLDH scale’s concurrent validity. All of the correlations between the variables were significant, indicating that the scale had a good concurrent validity (see Table 4). Furthermore, the correlations between CLDH with CA, CE, SC, and SI were 0.666 (*p* < 0.001), 0.477 (*p* < 0.001), 0.475 (*p* < 0.001), and 0.370 (*p* < 0.001), respectively, which fell well below the criterion of 0.700 to claim discriminant validity [80].

## 5. Discussion

The Children’s Hope Scale is one of the most widely used measures of child hope and has been translated into several languages for use across various cultures [35,82]. Nevertheless, there is a scarcity of validated measures with satisfactory concurrent validity, subgroup consistency, and reliability on hope for career and life development. Moreover, according to previous studies, the unique social context of Hong Kong calls for a rigorous measurement instrument to help young people identify and develop their hope for career and life pursuits. In this context, the present study attempted to contribute to the literature by developing and validating a two-factor structure (i.e., career and life development agency as well as career and life development pathways) of the CLDH scale among NEY who participated in the project titled CLAP@JC in the Hong Kong context.

The findings of this study showed the applicability of the CLDH scale among NEY in Hong Kong. The PCA carried out on the 20-item CLDH scale suggested a two-factor structure, namely CLDP and CLDA. The CLDH scale’s two-factor structure accounts for 63.08% of the variance. Furthermore, the results of the CFA supported the two-component constructs obtained from the EFA. Moreover, the CFA results show satisfactory fit indices, and all items in each subscale significantly represent their corresponding sub-construct, indicating that the CLDH scale has a construct validity. In addition, a subsample consistency analysis was also performed across the subgroups of gender, age, and years of residence in Hong Kong, and the results of this analysis showed a good model fit across the pairs of each subgroup. Thus, the findings show subgroup consistency and confirm the CLDH scale as a promising measurement tool.

The overall results reveal that the CLDH scale demonstrated adequate reliability as well as a satisfactory convergent validity, discriminant validity, and concurrent validity. To determine the reliability of the CLDH scale, the internal consistency was calculated. The reliability coefficients of the total scale and subscales were adequate, as indicated by the high Cronbach’s alphas (ranging from 0.892 to 0.950) and high composite reliability coefficients (ranging from 0.828 to 0.950), thus easily surpassing the standard that a scale’s reliability is sufficient if its reliability coefficient exceeds 0.700 [79,81]. The CLDH scale also showed a satisfactory convergent validity in the high correlation between CLDP and CLDA as well as satisfactory discriminant validity as indicated by the significantly positive yet moderate correlations between CLDH with CA, CE, SC, and SI (correlation coefficients ranging from 0.370 to 0.666; *p* < 0.001). Furthermore, the CLDH scale and its subscales were found to be positively correlated with both career-related outcomes and social well-being outcomes. According to the literature, hope for career and life development is positively related to career adaptability [57], youth career development competency [28], civic engagement [65], social contribution [64], and social integration [83]. The significant associations in the expected directions found in this study support the CLDH scale’s strong concurrent validity. All of the results show that the scale has the sufficient components for assessing NEY’s hope for career and life development in the Chinese context of Hong Kong society.

In summary, the present study has several theoretical contributions. First, despite the increasing recognition of the significance of hope in predicting various developmental outcomes [16,17], there is a scarcity of research exploring what hope means in the context of career and life development and how it is measured among NEY, a particularly vulnerable youth group undergoing various challenges and uncertainties during their career and life development processes [43,47]. This study is a pioneering attempt to develop a valid and reliable measurement instrument that can be used among NEY to understand their status of career and life development hope. Built upon Snyder’s [15] conceptualization of hope, we developed the 20-item CLDH scale with two key components: career and life development pathways and career and life development agency. Moreover, our findings from the EFA show that the final two-factor solution of the 20 items on the CLDH scale accounted for a significant proportion of the variance in NEY’s hope for career and life development. In this connection, this study complements existing studies [34] by confirming that the two key components (i.e., pathways and agency) suggested in Snyder’s [15] hope model are inclusive and applicable in the context of CLDH among NEY. In addition, this study also demonstrated the good concurrent validity and subgroup consistency of the CLDH scale, both seldom examined in relevant validation studies, and confirmed that the newly developed scale is valid and stable across gender, age, and residence-year subsamples.

Second, while there are unique sociocultural characteristics of the Hong Kong Chinese context that may complicate NEY’s CLDH [43,55], the results of this study provide empirical support for the applicability of the CLDH scale among Chinese NEY in Hong Kong. In particular, in hope-related literature, scholars have raised concerns about whether the agency component in Snyder’s [15] hope model, which emphasizes individual motivation, applies to the Chinese context [84,85], especially considering Chinese people typically receive motivation (i.e., agency) from others, such as parents’ affirmation, instead of their personal preferences [51]. As such, our results of both the EFA and CFA contribute to the research agenda by demonstrating that agency acts as a critical component when conceptualizing CLDH among Chinese NEY in Hong Kong and accounts for a large amount of the total variance (i.e., 23.14%). One possible explanation for this finding may be due to Hong Kong’s unique sociocultural context as a Westernized metropolitan city, which firmly adheres to traditional Chinese culture and is increasingly influenced by Western values simultaneously [78,86]. Thus, it is plausible that, with greater sustained exposure to Western cultures, today’s young people in Hong Kong are more likely to be oriented toward self-directed beliefs instead of striving for their parents’ affirmation in initiating and executing movements towards goals for their career and life development. In this way, although the collectivistic culture and family orientation in Chinese tradition are generally reported to hamper young people’s self-motivation regarding exploring their career and life development [43,55], the results of our study echo and yield empirical support for an emerging view that young people in Hong Kong might increasingly become more autonomous and independent, which is different from their counterparts in other Chinese societies [85].

On a practical level, the development of the CLDH scale also has several implications for career counselors, social workers, teachers, and NEY. First, given that current uncertainties surrounding the labor market have intensified the challenges of NEY’s career and life development [1,43], and therefore may further increase their level of hopelessness [9], our study regarding developing and validating the psychometric qualities of the CLDH scale suggests that the CLDH scale may be used by concerned professionals to assess the status of NEY’s hope in career and life development. Moreover, NEY might have unique needs or characteristics in career and life development compared to the general youth population [9,44], especially regarding how NEY initiate motivation for pursuing career goals from a motivation perspective (i.e., agency component) or how NEY generate routes to achieve their targets in a cognition–behavior context (i.e., pathways component). As such, the CLDH scale has the potential for integration into the needs assessment of career counseling practice, in which NEY’s career and life development pathways and career and life development agency could be rigorously explored by career counselors and social workers with the CLDH scale acting as a measurement tool. Moreover, career counselors, social workers, or teachers could utilize the CLDH scale to measure NEY’s changes in the CLDH level after implementing relevant services. Professionals and their organizations could monitor, evaluate, and modify the service design based on the assessment results of the CLDH scale. Additionally, there are various pre-employment training schemes and programs for NEY in Hong Kong that attach importance to promotion and sharing of career or further education information [43] and tend to overlook NEY’s needs in career and life development hope. Thus, the CLDH scale could serve as a road map for relevant training programs in practice. It provides current career intervention programs with a tool to design the service component and support NEY to explore how their hope for career and life development, particularly when they are in the struggling process of navigating the transitions from education to employment.

Aside from the above-mentioned contributions and significance, some limitations need to be noted and addressed in future research. First, the current study sample consisted of NEY who participated in the CLAP@JC project. Although the CLAP@JC project is a territory-wide project in Hong Kong, the generalizability of our findings is still subject to scrutiny in the absence of a randomized representative sample. Considering the unique sociocultural characteristics of the Hong Kong Chinese context, it is recommended that future research replicate our study and validate whether the CLDH scale could be applied to another cultural context to further analyze its cross-context applicability. Similarly, while the purpose of developing the CLDH scale for facilitating NEY’s career and life development and the sample of this study consisted of all NEY, it is still possible that the CLDH scale would be reliable and valid among other groups of youth; therefore, future studies may examine the use of this scale among a wider diversity of young people. Moreover, our study exclusively used self-reported data to assess CLDH among NEY from their own perspectives, which might have allowed for a mono-method bias. In response, future studies might consider collecting and triangulating data from other sources, such as parents’ appraisals for assessing NEY’s CLDH level and career counselors’ ratings of NEY’s CLDH in order to verify our newly developed scale. In addition, as the present study is cross-sectional, future studies might consider collecting longitudinal data and further verifying the test–retest reliability of the CLDH scale.

## Figures and Tables

**Figure 1 ijerph-19-10283-f001:**
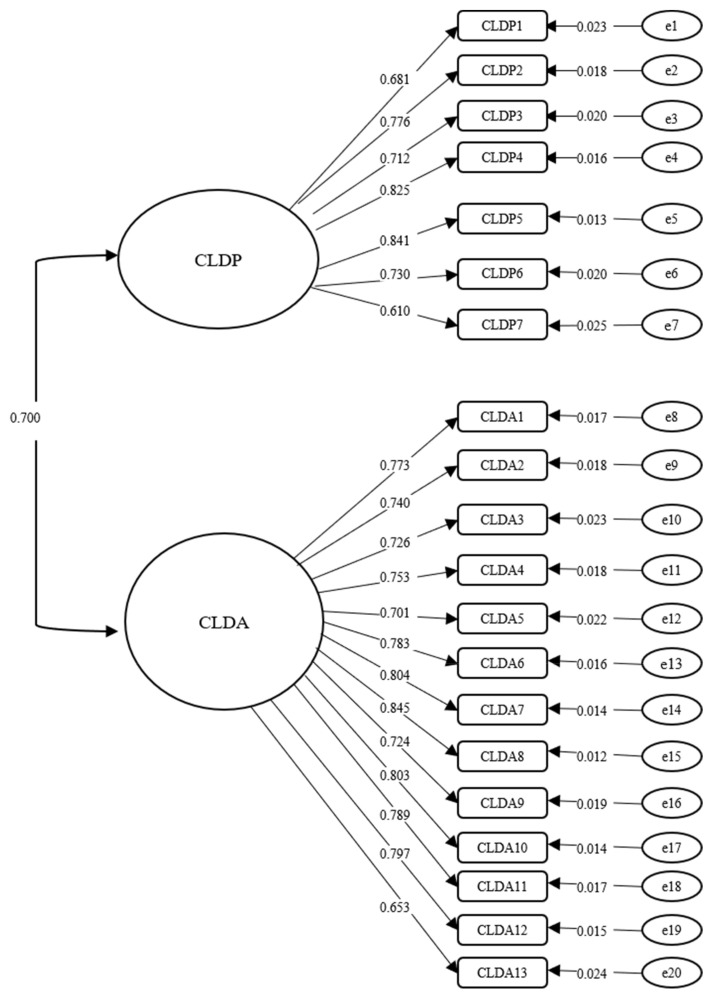
CFA factor structure and the standardized factor loadings (*n* = 1018). CLDP = career and life development pathways; CLDA = career and life development agency. All coefficients shown in this figure are factor loadings that are statistically significant at the *p* < 0.001 level.

**Table 1 ijerph-19-10283-t001:** Rotated factor loadings matrix from EFA (*n* = 980).

	Items	Factors	
		F1	F2
CLDP1	Engaged in education or training.	0.736	
CLDP2	Participated in activities to enhance self-understanding and career interests.	0.778	
CLDP3	Had encounters with business sector to understand the industry.	0.785	
CLDP4	Took initiatives to set career and life development goals and plans.	0.723	
CLDP5	Had a clear direction and meaningful engagements (e.g., education, employment, or training) in career and life development.	0.730	
CLDP6	Took actions to overcome barriers and difficulties.	0.592	
CLDP7	Took initiatives to launch youth-led activities.	0.720	
CLDA1	Was confident in establishing career roadmap.		0.743
CLDA2	Understood self-VASK (values, attitudes, skills, and knowledge), interests, and strengths.		0.733
CLDA3	Had career and life development aspirations.		0.738
CLDA4	Learned about different available careers and life development tools and resources.		0.708
CLDA5	Felt that future career and life development will be good.		0.737
CLDA6	Possessed the latest information about multiple pathways to inform study, leisure, and career options.		0.722
CLDA7	Was confident to make career choices that suit me.		0.779
CLDA8	Possessed knowledge, understanding and skills on career and life development.		0.805
CLDA9	Maintained positive work attitudes and life values.		0.772
CLDA10	Gained updated knowledge about multiple career and life development pathways.		0.765
CLDA11	Was motivated to pursue career and life development goals.		0.810
CLDA12	Effectively planned and managed career and life development.		0.762
CLDA13	Possessed resilience in facing difficulties.		0.705

Note: F1 = career and life development pathways (CLDP); F2 = career and life development agency (CLDA).

**Table 2 ijerph-19-10283-t002:** Scale statistics and item total correlations for the career and life development hope (CLDH) scale.

Subscale	Item	Scale Mean If Item Deleted	Scale Variance If Item Deleted	Item-Total Correlation	Cronbach’s Alpha for Subscale
Career and life development pathways (CLDP)	CLDP1	15.550	32.435	0.636	0.892
	CLDP2	15.307	31.737	0.745	
	CLDP3	15.790	32.834	0.695	
	CLDP4	15.263	32.517	0.741	
	CLDP5	15.295	31.781	0.758	
	CLDP6	15.050	33.897	0.649	
	CLDP7	15.791	33.968	0.614	
Career and life development agency (CLDA)	CLDA1	38.686	86.832	0.758	0.950
	CLDA2	38.538	87.196	0.731	
	CLDA3	38.396	87.355	0.720	
	CLDA4	38.555	86.941	0.737	
	CLDA5	38.394	87.786	0.716	
	CLDA6	38.550	87.285	0.747	
	CLDA7	38.453	85.979	0.771	
	CLDA8	38.540	86.326	0.817	
	CLDA9	38.285	87.155	0.731	
	CLDA10	38.611	86.740	0.778	
	CLDA11	38.328	86.330	0.791	
	CLDA12	38.646	87.154	0.783	
	CLDA13	38.368	89.364	0.657	

**Table 3 ijerph-19-10283-t003:** Factorial validation in subsamples according to gender, age, and years of residence in Hong Kong.

	CFA of Total Sample Model *N* = 1998	Gender		Age		Residence in Hong Kong	
		Male *n* = 1036	Female *n* = 962	Younger *n* = 1104	Older *n* = 894	Shorter *n* = 996	Longer *n* =1002
Chi-square	1335.775	686.365	853.160	823.447	696.020	742.485	763.468
Degrees of freedom	169	169	169	169	169	169	169
*p*-value	*p* < 0.001	*p* < 0.001	*p* < 0.001	*p* < 0.001	*p* < 0.001	*p <* 0.001	*p* < 0.001
CFI	0.938	0.946	0.929	0.933	0.943	0.945	0.932
RMSEA	0.059	0.054	0.065	0.059	0.059	0.058	0.059
SRMR	0.038	0.036	0.044	0.041	0.039	0.039	0.041

**Table 4 ijerph-19-10283-t004:** Correlations between CLDH scale and its subscales with career-related and social well-being outcomes.

	1	2	3	4	5	6	7	8
1. CLDH	1.00							
2. CLDP	0.859 ***							
3. CLDA	0.942 ***	0.636 ***						
4. CA	0.666 ***	0.477 ***	0.690 ***					
5. YCDC	0.719 ***	0.538 ***	0.729 ***	0.730 ***				
6. CE	0.477 ***	0.497 ***	0.392 ***	0.326 ***	0.424 ***			
7. SC	0.475 ***	0.483 ***	0.398 ***	0.369 ***	0.465 ***	0.826 ***		
8. SI	0.370 ***	0.264 ***	0.384 ***	0.465 ***	0.494 ***	0.243 ***	0.329 ***	1.00

Note: CLDH = career and life development hope; CLDP = career and life development pathways; CLDA = career and life development agency; CA = career adaptability; YCDC = youth career development competency; CE = civic engagement; SC = social contribution; SI = social integration. *** *p* < 0.001.

## Data Availability

The datasets generated and/or analyzed in the current study are not publicly available, as they contain information that could compromise the privacy of research participants. The data that support the findings of this study are available from the corresponding author upon reasonable request.

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
