# Peer review of "Development and Validation of the Career and Life Development Hope (CLDH) Scale among Non-Engaged Youth in Hong Kong"

_ijerph, 2022, doi:10.3390/ijerph191610283_

Round 1

Reviewer 1 Report

Thank you for inviting me to review the paper entitled “Development and Validation of the Career and Life Development Hope (CLDH) Scale among Non-engaged Youth in Hong Kong”.

Study is well written and designed; methodologically sound.

 Some minor recommendations are provided:

What is the language used for the survey? If Chinese is used, are the Chinese version of the scale validated? Back translation?

Normality of the data? How about the composite reliability, convergent validity and discriminant validity? Bootstrapping?

Provide further elaboration or discussion with regards to CLDP and CLDA.

As noted in previous studies, hope is a positive expectation of goal attainment and likely be able to explain through the combination of goal oriented/directed planning – pathways CLDP and motivation/agency CLDA. Looking into the items, they are also quite similar to self-efficacy – please clarify.

 In sum, the paper is well written, just some minor clarification needed.

Reviewer 2 Report

Thank you for the opportunity to revise this paper titled “Development and Validation of the Career and Life Development Hope (CLDH) Scale among Non-engaged Youth in Hong Kong”

Overall I think the paper is well written and deals with an interesting and relevant topic. I have, however, a few comments for the authors

First of all, I would recommend the authors to number the first section as Introduction, and the next one as Theoretical Background (or something similar) as the second section, as this is more common in the literature, rather than the theoretical section as a subsection of the introduction.

Also, current section 1.2.4 could be better fit in the introduction, to clearly state the goal f the paper, rather than in the theoretical section.

What was the language of the questionnaire? Were all participants native in the language?

Did you check for potential early versus late response? And non-response bias?

Was it possible to minimize potential common method bias by triangulating any variable/data with other sources?

Good luck with your research!

Author Response

Point 1: Thank you for the opportunity to revise this paper titled “Development and Validation of the Career and Life Development Hope (CLDH) Scale among Non-engaged Youth in Hong Kong” Overall I think the paper is well written and deals with an interesting and relevant topic. I have, however, a few comments for the authors
Response 1: Thank you very much for taking the time to review our research and also for your positive feedback! We have carefully read your valuable suggestions and revised the manuscript accordingly, with changes highlighted in yellow in the following responses and the revised manuscript.

Point 2: First of all, I would recommend the authors to number the first section as Introduction, and the next one as Theoretical Background (or something similar) as the second section, as this is more common in the literature, rather than the theoretical section as a subsection of the introduction. Also, current section 1.2.4 could be better fit in the introduction, to clearly state the goal f the paper, rather than in the theoretical section.

Response 2: Many thanks for your valuable suggestions. First, we have now separated the content related to the theoretical background from Section 1. Introduction and relabeled it as Section 2. Literature Review, which contains Section 2.1. Conceptualization of Hope and Theoretical Basis and Section 2.2. Measurement Gaps (Lines 80–208 on Pages 2-5). Second, we agree that it would contribute to smoothing the rationale of our research purpose if Section 1.2.4. Current Study in our previous manuscript were incorporated in Section 1. Introduction. Thus, the manuscript has been adjusted accordingly (Lines 74–79 on Page 2).

Point 3: What was the language of the questionnaire? Were all participants native in the language?

Response 3: Thank you for your questions. First, both Chinese and English were used in the survey to enable Chinese-speaking and non-Chinese-speaking participants to respond. The Chinese version was translated from English and was validated through back-translation procedures. We added the back-translation procedures to Section 3.2. Measures. The information is also presented below (Lines 242–244 on Page 5):

“The CLDH scale was translated into Chinese. The semantic equivalence of the translated Chinese version to the original scale was verified through back-translation procedures.”

Second, regarding the participants, 89.2% were Chinese and 10.8% were South or Southeast Asian minorities, as presented in the Procedure and Participants section. Therefore, an English-language version was provided for non-Chinese-speaking participants.

Point 4: Did you check for potential early versus late response? And non-response bias?

Response 4: Many thanks for your questions. First, data for this validation study were collected immediately after non-engaged youth (NEY) applied to participate in the CLAP@JC project. In other words, although the CLAP@JC is an evolving project, both early and late respondents were invited to complete the questionnaire at the time they participated in the project. Second, all the participants accepted our invitation to complete the questionnaire (response rate = 100%), thereby preventing our results from being affected by the potential non-response bias.

Point 5: Was it possible to minimize potential common method bias by triangulating any variable/data with other sources?

Response 5: Thank you very much for your instructive question! We agree that it is possible that when the present study used NEY’s self-reported data to validate the newly developed CLDH measure, it may have increased the possibility of introducing a mono-method bias. Hence, we have added this as one of the limitations of our study and further elaborated on it in Section 5. Discussion, as well as presenting it below (Lines 544–548 on Page 13):

“Moreover, our study used the self-reported data exclusively to assess CLDH among NEY from their own perspectives, which might have allowed a mono-method bias. In response, future studies might consider collecting and triangulating data from other sources, such as parents’ appraisals for assessing NEY’s CLDH level and career counselors’ ratings of NEY’s CLDH in order to verify our newly developed scale.”

Round 2

Reviewer 2 Report

No further comments